# Detection of Polonium-210 in Environmental, Biological and Food Samples: A Review

**DOI:** 10.3390/molecules28176268

**Published:** 2023-08-27

**Authors:** Lei Zhou, Rui Wang, Hong Ren, Peng Wang, Yiyao Cao

**Affiliations:** 1Department of Occupational Health and Radiation Protection, Zhejiang Provincial Center for Disease Control and Prevention, Hangzhou 310051, China; zhoulei@cdc.zj.cn (L.Z.); w9906070012@163.com (R.W.); hren@cdc.zj.cn (H.R.); pwang@cdc.zj.cn (P.W.); 2School of Public Health, Suzhou Medical College, Soochow University, Suzhou 215123, China

**Keywords:** ^210^Po, detection, environmental samples, food, biological samples

## Abstract

Ingestion of polonium-210 from environmental media and food can cause serious health hazards (e.g., gastrointestinal symptoms, tumours, etc.) and has been a public health concern worldwide since the 2006 poisoning of Agent Litvinenko ^210^Po in Russia. With the development of uranium mining and applications of nuclear technology in recent decades, the radioactive hazards posed by ^210^Po to living organisms and the environment have become increasingly prominent. In order to strengthen the monitoring of environmental ^210^Po and protect both the environment and human health, a series of explorations on the methods of ^210^Po determination have been ongoing by researchers across the globe. However, previous reviews have focused on individual sample types and have not provided a comprehensive account of environmental, food, and biological samples that are closely related to human health. In this work, the sources, health hazards, chemical purification, and detection methods of trace ^210^Po in different sample types are systematically reviewed. In particular, the advantages and disadvantages of various pretreatment methods are compared, and relevant domestic and international standards are integrated, which puts forward a new direction for the subsequent establishment of rapid, simple, and efficient measurement methods.

## 1. Introduction

### 1.1. Properties of ^210^Po

Polonium is a metallic element with an atomic number of 84 that was discovered by the Curies in 1898 when processing uranium ore [1,2]. The element is soluble in concentrated sulfuric acid, nitric acid, and diluted hydrochloric acid, as well as several other solutions, and it can also form soluble salts with chloride, nitrate, and other inorganic anions. Polonium has 43 isotopes [3] that range from 186 to 227 relative atomic masses, and all of them are radioactive, though ^208^Po, ^209^Po, and ^210^Po are the most stable. Other isotopes have short half-lives. ^210^Po the most widely used isotope [4] and is a radionuclide with high chemical and radiological toxicity [5].

In nature, ^210^Po is an alpha radionuclide derived from the natural radionuclide ^238^U through a series of decays [6,7]. The main decay process occurs when^238^U undergoes a series of decays to ^210^Pb, ^210^Pb undergoes beta decay to ^210^Bi, ^210^Bi undergoes beta decay to ^210^Po, and finally ^210^Po undergoes alpha decay to produce the stable 2^06^Pb (Figure 1) [6]. Polonium-210 has an alpha particle energy of 5.38 MeV and a half-life of 138.4 d [8], which is the longest half-life of all naturally occurring polonium radionuclides. It has high specific activity (1.66 × 10^14^ Bq/g) [9] and high volatility [10] and can easily form colloids and adsorb on metal surfaces. 

### 1.2. Sources of ^210^Po

The sources of ^210^Po in the environment are divided into two main categories: natural and artificial sources. During anthropogenic activities such as uranium mining and hydrometallurgical processing, important decay substrates of the natural uranium system, such as ^210^Pb and ^210^Po are discharged with wastewater into the surface water and soil around mines, causing pollution of the hydrosphere and soil lithosphere [2]. Airborne ^210^Po comes mainly from natural factors such as the decay of ^222^Rn on the surface of the earth’s crust [12,13], resuspension and wind erosion of soils [14], and volcanic eruptions [15]. Human activities such as fossil fuel combustion, phosphoric acid production, and fertilizer use also increase atmospheric ^210^Po levels [16]. Atmospheric ^210^Po and ^210^Pb are rapidly adsorbed onto aerosol particles [17] and can migrate through aerosols over long distances and for long periods of time, eventually entering the water or settling to the soil surface as dry or wet depositions. Soil and atmospheric ^210^Po can also enter plant tissues by root uptake or surface adsorption. Eventually, the radionuclide ^210^Po can become enriched in the human body via our natural food chain (Figure 2).

### 1.3. Health Risks of ^210^Po

Currently, ^210^Po has been found to be widely distributed in a variety of environmental and biological samples [1,18,19], such as groundwater [20,21,22], fish, and shellfish [20,23,24]. ^210^Po in environmental media can enter the body through inhalation, diet, drinking water, open wounds, and the food chain [17,25]. In fact, a significant proportion of an individual’s exposure to natural background radiation through ingestion of food and water comes from the radionuclide ^210^Po [26], which provides approximately 60% of the annual effective dose of all naturally ingested radionuclides [25].

Fifty to ninety percent of the ^210^Po that enters the body rapidly reaches the gastrointestinal tract [1]. After uptake by the gastrointestinal tract, ^210^Po is mainly bound and concentrated in red blood cells and plasma proteins [27]. The remainder reaches various organs via the bloodstream, is retained in the bones, lungs, kidneys, and liver, and is excreted in the form of excrement, urine, or sweat [6]. Because the alpha particles released by ^210^Po during decay are weak and easily shielded by the skin, the harm caused by external exposure to ^210^Po is negligible. However, once inside the body, it is extremely harmful, causing strong ionization in the body, destroying the genetic material of human cells, and triggering a series of serious biological effects [27]. Therefore, it is of great public health significance to enhance the monitoring of ^210^Po radioactivity levels in environmental media to prevent and mitigate ^210^Po hazards to both human health and the natural environment.

### 1.4. Analytical Method Reviews for ^210^Po

In the 21st century, a large number of scholars have conducted research on ^210^Po, especially after 2011; the number of research results on the keyword “^210^Po” on the Web of Science has doubled compared with previous years. The reason for this surge in research may be related to the Fukushima nuclear accident in 2011 (Figure 3). At present, however, there is no coherent review of the ^210^Po detection literature. To the best of our knowledge, only Matthews [6] and Thakur et al. [28] have reviewed the determination methods of ^210^Po in environmental samples. But there is still a lack of review of ^210^Po determination methods in food samples, air samples, and biological samples, as well as a better outlook and prospective prediction of future analytical methods for ^210^Po. A review of the ^210^Po method needs to cover a wide range of different sample types.

In this paper, we systematically summarize the pretreatment, chemical separation and purification, preparation, and detection methods of ^210^Po in different environmental media, biological samples, and seafood. Table 1 presents the methods reported since the start of the 21st century for the detection of ^210^Po in environmental and biological samples. In general, the analytical procedure for determining whether ^210^Po is present in a sample is divided into four stages: pretreatment, chemical purification, deposition source preparation, and measurement, as shown in Figure 4 and described in detail later in the paper.

## 2. Sample Pretreatment

Due to the variety of sample types and quantities, different pretreatment methods are often used for ^210^Po testing depending on the specific sample type in question. The most crucial aspect of pretreatment is temperature control, which prevents polonium from volatilizing at high temperatures and resulting in a significantly reduced radiochemical yield [2,48]. Relevant studies have shown that various polonium compounds begin to volatilize at 100 °C, and when the temperature reaches 300 °C, 90% of the polonium can completely evaporate [10]. Chlorides, organic compounds, and chelates of polonium are particularly volatile, and all polonium complexes volatilize below 200 °C [49]. Consequently, temperature control is essential during the pretreatment of both environmental and biological samples. In addition, the time required for processing can be reduced by different pretreatment methods while achieving high yields.

### 2.1. Pretreatment of Environmental Samples

#### 2.1.1. Pretreatment of Water Samples

Due to the relatively low levels of ^210^Po in natural environmental water samples, the volumes required to measure ^210^Po in water samples are usually large [50], so enrichment and concentration prior to testing are required. Currently, the treatment of water samples is often carried out using evaporation, co-precipitation, and chelation methods [26]. Due to its simple operation process when processing large-volume water samples, the direct evaporation method is preferred for concentrated environmental water samples [20,51]. However, evaporation is extremely time-consuming when processing large volumes of samples (>1 L), so the method is not suitable for radiological emergencies involving ^210^Po or samples containing large amounts of dissolved impurities [52]. 

The co-precipitation method is widely used to concentrate and enrich polonium in water samples due to the fact that it can save time compared with other methods, involves non-volatilization, and can remove interfering ions [6,28]. At present, the most popular methods for enriching ^210^Po in environmental water samples include iron hydroxide [Fe(OH)_3_] co-precipitation [22,53,54], manganese dioxide (MnO_2_) co-precipitation [55,56,57], and calcium phosphate [Ca_3_(PO_4_)_2_] co-precipitation [45].

Iron hydroxide [Fe(OH)_3_] co-precipitation, which involves first adding ^209^Po tracer to the water sample, uses iron hydroxide as a carrier to adsorb the carrier water ^210^Po and ^209^Po, dissolving them both in hydrochloric acid, and then adding ascorbic acid to reduce Fe^3+^ to Fe^2+^, followed by self-deposition for source production. When iron hydroxide is used for preconcentration, the solvent extraction step must be performed using an extractant such as diisopropyl ether to remove iron ions from the sample solution [58], as large amounts of iron ions can interfere with solvent extraction, affect the purification of polonium by extraction chromatography, and even affect the auto-deposition of polonium.

Due to the cumbersome nature of iron hydroxide co-precipitation and its use of toxic or flammable organic reagents, researchers have explored other methods of preconcentration [59,60]. Co-precipitation using MnO_2_ is widely used since with this method no interfering ions are present, and the method is particularly suitable for the pre-enrichment of polonium in large volumes (>1 L) of water samples [28]. Lee et al. [52] concentrated polonium nuclides with manganese dioxide precipitation and purified them by solvent extraction. ^209^Po tracer was added to their tap water sample and stirred well. Then, KMnO_4_ and MnCl_2_ were added, and the solution was adjusted to a pH of 9 with 25% NH_4_OH. The sample solution processed by this method could then be directly used for radiochemical separation. In contrast to the traditional MnO_2_ co-precipitation method, their method required no evaporation step, thus saving a lot of experimental time.

The calcium phosphate co-precipitation method first adds calcium nitrate and ammonium biphosphate to an aqueous sample, adjusts the pH to 10 with ammonium hydroxide, and then forms a co-precipitation with the resulting calcium phosphate [61]. Maxwell et al. [45] used a new rapid method developed at the Savannah River National Laboratory (SRNL) to measure ^210^Po in water samples in which they used rapid calcium phosphate co-precipitation for sample pretreatment, separation, and purification using *N,N,N′,N′*-tetra-n-octyldiglycolamide (DGA) resin, and then micro-precipitation of ^210^Po using bismuth phosphate that they counted using alpha spectrometry. This new method allows rapid detection in a short time with excellent removal of interfering ions, and its high chemical yield (>95%) and excellent alpha peak resolution make it suitable for both emergencies and routine water analysis. ^210^Po in water can also be precipitated by a chelating agent (ammonium cobalt dithiopyrrolidine disulfate, CO-APDC), which has been shown to be a prerequisite not only for quantitative extraction of polonium from samples [26] but also for direct extraction of polonium from water samples [26,62,63,64].

In summary, the co-precipitation method is prized by a wide range of researchers for its ability to pretreat large volumes of water samples. Its advantages are primarily time-saving, much less loss of polonium due to volatilization than evaporation, and the strong ability to remove interfering ions. Nonetheless, co-precipitation is labour-intensive as the treatment steps are cumbersome and require constant supervision by the experimenter.

#### 2.1.2. Pretreatment of Air Samples

The range of activity concentrations in ground-level air for ^210^Po is 0.03–0.3 Bq/m^3^ [13], and the background radiation received by inhalation of ^210^Po accounts for more than 70% of the average total dose of radiation to the human body [65]. ^210^Po entering the body poses considerable health risks by damaging the internal structure of the body and thereby increasing the risk of cancer. However, there is currently no national standard for the determination of ^210^Po in the air in China. The main challenge for the detection of ^210^Po in air samples is the question of how to separate ^210^Po from aerosols and establish a hydrochloric acid self-deposition system [65]. The most prominent existing research idea on this subject is to first convert the ^210^Po in the sample into Po^2+^ or Po^4+^ in the particle state [66], digest and destroy the organic matter adsorbed in the aerosol with concentrated nitric acid and concentrated sulfuric acid, and then leach it several times with concentrated hydrochloric acid to form a hydrochloric acid system where self-deposition can occur. 

However, the high-temperature acid digestion methods normally used for many solid samples can result in the loss of polonium through volatilization unless carried out in specially designed containers such as Kjeldahl flasks [67]. Khaing et al. [68] used alkaline melting to treat ^210^Po in air filter samples, where the main step was to dissolve glass or cellulose air filters with sodium hydroxide and hydrogen peroxide, followed by extraction chromatography to separate ^210^Po. Unfortunately, the recovery yield of this method was low (<60%), indicating that there was still organic material that was fully digested. Maxwell et al. [32] therefore investigated a new method for recovery and measurement of ^210^Po on air filters after fusion of the air filter with sodium nitrate or potassium nitrate and sodium hydroxide (1:1). In a rapid sample, polonium was rapidly separated using iron hydroxide co-precipitation and DGA resin separation. This method has the advantages of high chemical yields and effective removal of sample matrix interference [32].

In addition to being present in aerosols, ^210^Po can also be found in the fumes produced by burning tobacco. Studies have shown that more than half of the radioactive substances emitted by burning cigarettes are released into the air, and even nonsmokers (second-hand smokers) can inhale these substances [69]. The smoke inhaled by smokers transports about 32% of the ^210^Po in each cigarette to the smoker’s lungs [70,71] and adsorbs it in the bronchi, where it is not easy to remove and poses a hidden danger to health [69]. In the past decade, many scholars have studied radionuclides in tobacco, especially the exploration of ^210^Po [72,73,74]. One group ground tobacco taken from cigarettes into a homogeneous powder and acid digested it with HNO_3_, HClO_4_, and HF [73]. Berthet et al. [75] washed ^210^Po from flue gas with three consecutive 1 M HCl wash bottles, F1 to F3. Independent analysis of each flask fraction showed that F1 retained 70% of the total ^210^Po activity in the flue gas of conventional cigarettes, F2 retained about 20%, and F3 retained about 10%. These results show that the acid digestion method reduced the volatilization of polonium caused by high-temperature combustion and better retained ^210^Po in tobacco and smoke.

#### 2.1.3. Pretreatment of Solid Samples

For solid environmental samples such as sediment, soil, and sludge, the pre-treatment process includes both physical and chemical treatment. The samples are first dried, ground, and sieved to obtain a homogenized sample. Chemical treatment involves the dissolution of the sample itself as well as the destruction of organic matter and is usually achieved by any of three methods: wet ashing in an open system, acid digestion in a pressure vessel, or microwave digestion [6]. The traditional wet ashing method uses a mixture of HCl, HF, HNO_3_, and HClO_4_ in different proportions, heated at different temperatures for different times, to destroy the organic material in the sample. This method has several disadvantages, however, such as time-consuming steps and the risk of external sample contamination, the formation of insoluble salts during sample evaporation, and the loss of polonium from volatilization during heating [76].

The melting method is also commonly used for the complete dissolution of solid samples (e.g., soils, sediments, rocks, and solid biological samples containing refractory matrices) [28]. It is carried out by heating the sample with various molten mixtures (e.g., hydroxides, peroxides, carbonates, bisulphates, pyrosulphates, alkali metal borates, fluoride-pyrosulphates, etc.) in graphite, nickel, zirconium, or platinum crucibles [32,35] above a burner or in a muffle furnace until the molten mixture is completely melted and clarified. After cooling, the molten mass is dissolved in diluted nitric or hydrochloric acid. This method is more aggressive than acid digestion and is able to dissolve insoluble substances such as silicates, resulting in a more homogeneous and pure solution that contains ^210^Po.

### 2.2. Pretreatment of Biological Samples

In radiological monitoring and risk assessment, biological samples such as urine, blood, and hair are often analyzed for ^210^Po. For such samples, similar pre-treatment methods to those used for environmental samples can be used, with the appropriate method selection depending on the sample type. Urine is a complex matrix that contains many organic compounds and some inorganic salts [77]. The main problem with urine samples is the possible presence of suspended particles that may be retained by filters, resulting in poor recovery and/or spectral resolution [21]. To combat this, Manickam et al. [1] used a combination of concentrated nitric and hydrochloric acids for the acid digestion of urine. However, the digestion was not allowed to dry, thus avoiding polonium loss due to volatilization and greatly reducing the total sample preparation time. After cooling, concentrated NH_4_OH was added to raise the pH to approximately 9. MnCl_2_ and KMnO_4_ were then added, and Po was co-precipitated with the formed MnO_2_. In emergency situations, this method can prepare and measure samples within 24 h. In another study, Guérin et al. [21] used KBrO_3_ to oxidize and filter urine samples and found that samples oxidized with KBrO_3_ were filtered faster than those without the oxidant.

Both blood and hair are pretreated with a wet digestion process similar to urine. The organic matter in the blood is destroyed with a mixture of concentrated nitric acid [78] or 1:1 concentrated nitric acid and 20% H_2_O_2_ oxidation, and polonium ions are released into the solution [79]. After evaporation, the dry residue is dissolved in dilute hydrochloric acid for subsequent treatment [80]. For the pretreatment of hair samples, the main difference is that the collected hair is repeatedly washed with detergent so that it does not contain foreign matter. Finally, the hair is rinsed with distilled water, dried for 4–5 h, and stored in polyethylene bags for further processing [81].

### 2.3. Pretreatment of Food Samples

With its surface particle chemistry and high affinity for proteins [9,82], ^210^Po is readily adsorbed to the surface of phytoplankton in the marine environment [4] and is further transferred to subsequent trophic levels. Protein-rich organisms, therefore, typically have high concentrations of ^210^Po activity [83]. Since ingestion is the most important route of ^210^Po entry into the human body, fish products are the main contributors to ^210^Po dietary intake [18]. The trophic levels of the aquatic food chain have a strong bioconcentration and biomagnification effect on ^210^Po, especially in marine organisms, which makes high-protein seafood the main source of human ^210^Po [79]. 

Currently, there are three methods of dissolution for biological samples: dry digestion, wet digestion, and microwave dissolution [84]. For sample dissolution, the wet digestion method has significant advantages over the traditional dry digestion method. First, the wet ashing method can effectively dissolve impurities in the sample and can operate at a lower temperature, thus increasing the efficiency of the process. In addition, the wet digestion method avoids the losses associated with the volatilization of polonium due to heating in the traditional dry ashing method, thus ensuring the accuracy of the sample. However, the efficiency of wet digestion is much lower than that of microwave digestion for samples that contain high amounts of oils [18].

The vast majority of analyses of ^210^Po in aquatic products in China are currently carried out by wet ashing [85,86]. The wet ashing method involves the addition of strong oxidants (e.g., concentrated nitric acid, concentrated sulfuric acid, perchloric acid, potassium permanganate, hydrogen peroxide, etc.) to the sample and heating the sample to decompose and oxidize the organic material completely [87]. After this, the components to be measured are transformed into their ionic states within the digestion solution. Dong et al. [85] analyzed ^210^Po levels in typical seafood from the Yellow Sea waters of China and proposed a rapid disposal of collected seafood to avoid interference from other factors. Their samples had to be kept at a low temperature (<10 °C) at all times until pretreatment, when they were first thawed, at which time the edible parts were removed, cleaned, weighed, dried, and ground. Once ground into a homogeneous powder, the samples were placed in refrigeration pending analysis. Each biological sample was dried and then added to a standard solution of ^209^Po tracer of known activity, and the organic matter was destroyed using a ‘tri-acid wet ashing’ method with nitric acid, hydrogen peroxide, and perchloric acid. In this case, the authors advise that the final solution should be colourless and transparent when the organic matter in the sample is sufficiently digested [85], but the oil and grease components in aquatic samples are often difficult to digest thoroughly, and incomplete digestion can seriously reduce the efficiency of the ^210^Po self-deposition.

Microwave digestion is a pretreatment technique in which a substance sample is heated directly by both molecular polarization and ionic conductivity, causing the surface layer of the solid sample to break down rapidly, creating a new surface for the solvent to interact with, and allowing the sample to be completely digested within minutes [88]. Szarlowicz et al. [89] reported that for smaller masses (0.1–0.2 g) of sediment samples, microwave digestion using concentrated HNO_3_ + HCl gave acceptable recoveries of polonium and did not require the use of HF, effectively reducing both processing time and cost. Although the microwave digestion technique is a promising method, it is only capable of rapid and accurate detection of very small amounts of organic matter (typically 0.5 g dry, 2–3 g wet), so for larger masses of samples (20–25 g, wet weight), it may be necessary to split them into multiple digestion vessels to obtain the sensitivity required to measure low levels of ^210^Po [90]. 

In order to solve the above problems, Baki et al. [18] used the Digiprep HT250–10 digestion system for the pretreatment of fish. This system consists of a graphite heating block that accommodates ten tall cylindrical digestion tubes (300 mL, glass, 5 cm deep) that house heat to provide more uniform heating. The system can efficiently dissolve fish samples (20 g, wet weight) into a clear liquid solution in less than 7 h, and the digestion ends without any visible fat residue. Although microwave digestion technology such as this is characterized by simple operation, rapid and complete decomposition, low reagent consumption, low loss of volatile elements, and low blanking and is known as a “green chemical reaction technology” [67], the major drawback of this method is that it does not allow the processing of large numbers of samples (>5 g) at one time [58].

The advantages and disadvantages of all of the above pretreatment methods for environmental, biological, and food samples are detailed in Figure 5.

## 3. Chemical Purification

After sample pretreatment is completed, the source preparation of ^210^Po can be carried out directly. Direct deposition source preparation has the advantage of saving a lot of time in practical analysis, but for some environmental samples, direct deposition may be affected by other ions, resulting in a thicker deposition source or reduced Po recovery [6,91]. Depending on the deposition conditions, deposition of ^210^Pb and ^210^Bi may result when the sample has a high ^210^Pb/^210^Po and/or ^210^Bi/^210^Po ratio. Such samples need to be chemically separated and extracted first. The main chemical separation methods commonly used are discussed below and their advantages and disadvantages are summarized in Appendix A.

### 3.1. Solvent Extraction

Solvent extraction, also known as liquid-liquid extraction, is an effective method for separating, enriching, and extracting useful substances from solutions, and it is frequently used for the separation of radionuclides. The principle is to add an immiscible (or slightly immiscible) solvent to a liquid mixture and use the different solubilities of its components in the solvent to achieve separation or extraction [92]. Polonium has four stable chemical valence states, namely −2, +2, +4, and +6, with the +4 valence state being the most stable in most solutions [4] and in chloride solutions in the PoCl_6_^2−^ state [93] in particular. Using this property, polonium can be extracted from acidic solutions with a variety of extractants. Isopropyl ether, methyl isobutyl ketone, diisopropyl ketone, and tributyl phosphate can all be used to extract polonium from acidic solutions [6]. 

Diethylammonium diethyl dithiocarbamate (DDTC) has also been used to extract Po from HCl solutions into chlorinated hydrocarbon solutions (CHCl_3_, CCl_4_, or CH_3_CCl_3_) [91,94], and trioctylphosphine oxide (TOPO) is an excellent extractant for polonium in hydrochloric acid media that allows for essentially 100% of polonium to be extracted over a wide range of hydrochloric acid concentrations [95]. A major disadvantage of this method, however, is the generation of mixed radioactive waste during the separation process and the problem of third-phase formation [28].

### 3.2. Ion Exchange Chromatography

Ion exchange is a method of separating metal ions in solution by exchanging ions on an ion exchanger with ions in the solution [96]. The main process of this ion resin exchange includes six steps: preselection, pretreatment, column loading, resin exchange, resin elution, and resin regeneration [97]. Ion exchange chromatography is one of the most common methods used for ^210^Po purification and separation, and separation of polonium on anion exchange columns is usually performed in HCl media since polonium is strongly retained in 0.05–12 M HCl anion exchange resins such as Dowex-1, Dowex-2, and Bio-rad AG1-X4. Indeed, polonium has a high retention capacity in HCl media throughout the acidic range [28]. However, in HNO_3_ media, this retention capacity is very low. 

Interestingly, a method for the chemical separation of Au from fission products using anion exchange was developed by Douglas et al. This method demonstrated that Po could be eluted in the same fraction as Au with minimal interference from other elements using only a small volume of reagent [98]. This is more desirable than previous studies that used ion exchange and large elution volumes [99,100,101]. Moreover, the anion exchange resin Bio-Rad AG1-X4 has been shown to adsorb Po in acids of 0.1–12 mol/L strength [6]. Dowex anion resins have been used for the separation of Po, Pb and Bi, and cation exchange resins such as Bio-Rad AG50Wx8 can also effectively adsorb ^210^Po from dilute HF, HNO_3_, or H_2_SO_4_ and elute with a 7 mol/L HCl solution [6].

### 3.3. Extraction Chromatography

Extraction chromatography, also known as extractive lamination, is a new separation method developed in the 1970s after ion exchange and solvent extraction. The basic principle is the combination of liquid-liquid extraction and chromatography to achieve separation according to the different distribution ratios of the components in the two phases. This is a new separation technique for the separation of inorganic substances using an extractant adsorbed on an inert support or polymerized with a resin as the stationary phase and an inorganic aqueous solution as the mobile phase [102]. Vajda et al. [103] first reported a method for separating polonium using a crown ether column. This method selectively retains polonium and lead from a 2 M HCl solution using bis-4,4′(5′)-t-butyl-cyclohexano-18-crown-6-ether and then strips polonium with 6 M HNO_3_ and lead with a 6 M HCl solution. Recently, extractive chromatography has been increasingly used for the separation and purification of ^210^Po, with Sr-Spec resin in particular, which can retain Po from acidic solutions with a recovery of about 70% [103]. The retention capacity for polonium on Sr-resin is about 100 in 0.5–1.0 M HNO_3_. Above this concentration, the retention of Po (IV) drops off rapidly [103,104]. Therefore, one must pay special attention to the acid concentration when using Sr resin. 

DGA is another commonly used resin for separating polonium [105]. The presence of interfering ions such as Na^+^, K^+^, Mg^2+^, and Fe^3+^ in a sample has little effect on the retention of Po (IV) on DGA resin. However, the high cost of DGA is a major drawback to its use in conventional polonium analysis. Other solid extractants include extractive drench resins impregnated with di(2-ethylhexyl) phosphoric acid (HDEHP), which can be used to elute ^210^Po from dilute citric acid solutions [106]. Tri-n-butyl phosphate coated on a polytetrafluoroethylene support has also been found to extract Po well (100%) from a 6 mol/L hydrochloric acid solution but requires elution with a 2 mol/L HF solution [107]. 

In recent years, scholars have additionally proposed a new continuous extraction system for the continuous separation and enrichment of ^210^Po, uranium, and thorium isotopes. This system is composed of TRU resin, SR resin, and RA-01 and AG50Wx8 mixed columns (HRA resin) [42,43]. Compared with other methods, this new method reduces the processing time, amount of chemical reagent, and amount of resin needed [108], and it has already been applied to the separation and purification of water and slag samples.

Extraction chromatography combines the high selectivity of solvent extraction with the simple, multi-stage nature of column chromatography. The main advantages of this method are that the chromatographic resins have high selectivity for polonium, exchange kinetics faster than anion exchange, and are characterized by simple operation steps, easy sample processing, and low cost. More importantly, extraction chromatography can significantly reduce the volume of highly radioactive solutions and the amount of solid waste, as well as the harmful effects of radioactive waste on human health and the environment [54].

## 4. Source Preparation

The measurement of the radionuclide ^210^Po in water and food in HJ 813–2016 and GB 14883.5–2016 uses alpha spectrometry [109,110], which is one of the most popular methods in research. However, alpha spectrometry requires a homogeneous thin-layer source to obtain maximum resolution of the Po peak. Currently, there are three main methods for source preparation: self-deposition, electrodeposition, and micro-precipitation. We summarize the advantages and disadvantages of different methods to prepare sources in Appendix A.

### 4.1. Spontaneous Deposition

Due to its higher reduction potential than many metals, ^210^Po can undergo spontaneous electrochemical exchange with such metals [111]. Using this property, ^210^Po can be self-deposited onto metal surfaces to form a coating. Furthermore, ascorbic acid, sodium citrate, and hydroxylamine hydrochloride are used in the self-deposition process to reduce the effect of competing ions (e.g., Fe^3+^, Cr^6+^) present in some samples [28,112,113].

The self-deposited materials used in experiments are mainly silver flakes or silver foil made of metallic silver [114,115,116]. Due to the high cost of silver flakes, however, materials such as copper flakes [117], stainless steel flakes [81], and nickel flakes [118] have also been used instead of silver flakes. However, Li et al. [119] studied and compared these alternative self-deposited materials and concluded that silver flakes have better selectivity for Po, better resistance to other ions in solution, and better stability compared with copper flakes, for example. Another study of self-deposition found that the deposition rate of polonium on copper and nickel decreased as the concentration of hydrochloric acid increased, whereas the rate of self-deposition on silver did not change significantly with the concentration of hydrochloric acid [111]. The reason for this may be that copper and nickel are chemically more active than silver, and the higher the concentration of hydrochloric acid, the more the copper and nickel flakes will be corroded by the high concentration of hydrochloric acid, thus affecting the onset of self-deposition [50].

In terms of temperature selection for the self-deposition process in a water bath, Li et al. [119] found that the self-deposition recovery increased with increasing temperature and that the recovery tended to stabilize at a temperature of 70 °C to 96 °C, reaching 95.3% to 98.3%, which was in line with the ideal state. Du et al. [120] also found that the higher the temperature, the shorter the self-deposition time. When the self-deposition temperature was 80 °C and the self-deposition time was 2.5 h, the self-deposition time reached equilibrium and the α net count rate reached its maximum. Moreover, the self-deposition equilibrium time was maintained for a longer period. Karali’s research showed that at 0.5 M HCl, 70–75 °C for 3 h, the deposition efficiency of polonium on different metal discs could be ranked in the following order, from high to low: silver > nickel > stainless steel > copper [121]. Therefore, attention should be paid not only to the temperature of the water bath during the experiment to prevent the volatilization of ^210^Po due to high temperatures but also to the concentration of hydrochloric acid to avoid causing corrosion of the self-deposited metal.

### 4.2. Electrodeposition

Electrodeposition is also commonly used in the source production process for ^210^Po. In electrodeposition, polonium is electrochemically coated from an electrolyte solution onto a 10 mm diameter polished stainless steel disc for several hours at a constant current (250 mA) [122]. The mechanism of the electrodeposition process is controlled by Hansen’s theory of electrodeposition of lanthanide and actinide hydroxides at low current densities. High concentrations of hydroxyl ions are required near the cathode surface to precipitate hydroxides from very low mass concentrations of radionuclides in the electrolyte [2], and research has shown that direct electrodeposition of polonium from acidic solutions on carbon electrodes is 40–85% efficient [4]. Rieth et al. [123] studied the electrodeposition of ^210^Po in 0.1 M HNO_3_ solution on various electrode materials (Cu, Ag, Ti, Pd, and Ni in the form of 6 × 6 mm foils) and found that the highest deposition was achieved on Ni electrodes and the lowest on Pd electrodes. The electrodeposition results observed on Ni, Cu, and Ag were also satisfactory, indicating that they have potential in the preparation of ^210^Po source deposition as well.

### 4.3. Micro-Precipitation

In recent years, microprecipitation has become a desirable alternative to autodeposition for the preparation of polonium alpha counting sources. Compared with conventional autodeposition, micro-precipitation is much faster since it does not require heating and can therefore be used to process large sample volumes with high recoveries (80–95%) [28]. Although micro-precipitation can improve the yield of the autodeposition method in a shorter time without heating, it requires the chemical separation of Po from potentially interfering alpha-emitting nuclides [32]. Many studies have been carried out on different micro-precipitation methods for the preparation of ^210^Po counting sources, including copper sulphide [21], bismuth phosphate [124], and monolithic tellurium micro-precipitation [125].

#### 4.3.1. Copper Sulphide (CuS) Micro-Precipitation

Due to the low solubility of PoS, the α counting source of ^210^Po can be rapidly prepared in HCl solution by CuS micro-precipitation technology, and the final recovery rate of this method can reach about 85% when treating water and urine samples [21]. Compared with conventional plating, CuS micro-precipitation is faster, can more easily process large batches, and gives high recoveries (80–95%) without a heating step. However, the recoveries are significantly lower when the molar concentration of HCl is higher than 1 mol. Miura et al. [30] used copper sulphide micro-precipitation to separate ^210^Po from Sr resin and electro-deposited ^210^Po on stainless steel sheets. The recovery of this method ranged from 56% to 99%, with a large variation interval as well as instability. Therefore, the copper sulfide (CuS) micro-precipitation method may have only limited potential as the deposition source for the preparation of Po-210.

#### 4.3.2. Tellurium (Te) Micro-Precipitation

The tellurium micro-precipitation method uses hydrochloric acid to leach the polonium in the sample at a low temperature and uses the single Te precipitate produced by the reduction of Te by stannous chloride as a carrier to carry polonium. The α source of ^210^Po is then obtained by filtration, which was first proposed by Song et al. [125]. This method is faster than the traditional natural precipitation method and saves experimental time, while also allowing large sample volumes to be processed with >90% recovery throughout and without the heating step required for high recoveries, reducing the loss of ^210^Po due to heating and making the process both simpler and more efficient. Compared with copper sulphide micro-precipitation, tellurium micro-precipitation is more resistant to acidity and can better effect the rapid detection of ^210^Po in poorly soluble solid samples [126]. This is not only suitable for the measurement of polonium-210 in water samples but also for other environmental media such as soil and organisms [126]. However, it has been demonstrated that the reduction of Te(IV) to monomeric Te is incomplete in acidic systems in ascorbic acid and not in hydroxylamine hydrochloride. This phenomenon resulted in a very low recovery of polonium, almost close to zero [28].

#### 4.3.3. Bismuth Phosphate (BiPO_4_) Micro-Precipitation

The bismuth phosphate micro-precipitation method was established to separate ^210^Po in urine, specifically [124]. The procedure involves adding concentrated phosphoric acid to a well-digested urine sample to adjust the pH, followed by bismuth nitrate for bismuth phosphate precipitation. Since solutions containing inorganic salts precipitate an average of about 90% in 50 mL or 500 mL of urine, some researchers expect that this method can be applied to larger volumes of urine. However, although this technique shortens the preparation time of the counting source, it does not provide sufficient selectivity against other alpha radionuclides and also requires a purification step, making it more cumbersome than other methods.

## 5. Radioactivity Measurements

There are three main methods for the detection of ^210^Po in environmental and biological media: alpha spectrometry, liquid scintillation counting, and large-area screen grid spectrometry. The principles, advantages, and disadvantages of each method are summarized in detail in the following subsections and Appendix A.

### 5.1. Alpha Spectrometry

Alpha spectrometry using a silicon surface potential or PIPS detector has become the method of choice for the determination of ^210^Po in environmental and bioassay samples [28,84]. The main principle of this method is to add an isotopic tracer of known activity to a self-deposited system, count it using an alpha spectrometer after the counting source has been prepared, and calculate the activity concentration of ^210^Po from the count rate. The most commonly used tracers are ^208^Po and ^209^Po. ^208^Po is often chosen as the tracer due to its better practicality. However, in recent years, ^209^Po has been increasingly used because it has a longer half-life and its alpha ray energy is further away from that of ^210^Po [127], which means that there are more easily identifiable peaks when measuring alpha energy spectra. Unfortunately, ^209^Po tracers are difficult to obtain commercially and expensive, making them unsuitable for the detection of bulk samples [120]. Alpha spectrometry has the advantages of lower detection limits, higher energy resolution, and lower cost compared with other methods, but tracer contamination of the detection probe and the need for a complex pretreatment process are two of its major drawbacks.

In China, total α counting is an earlier method used in the national standard for the determination of ^210^Po in water. The total alpha counting method uses manganese dioxide and hydroxide as carriers to carry ^210^Po off from the sample, uses silver or copper flakes as carriers in an acidic solution to make the source, and then determines the total alpha activity in the sample as the activity of ^210^Po. This method is similar to the procedure described above for alpha spectroscopy but does not require the addition of polonium-210 isotopes for tracing. The full recovery of the method ranges from about 40% to 80%. The main advantage of total alpha counting is that it allows rapid self-deposition of the source and near-field measurements without tracer and without separation of matrix elements (high detection efficiency due to the close distance between source and detector), but it is more difficult to achieve a correction for full recovery for each sample. At present, the total alpha counting method is still used in the determination of ^210^Po radioactive substances in national standard foods. However, a ^210^Po standard solution was added to calculate the recovery rate in the detection process.

### 5.2. Liquid Scintillation Counting

Liquid scintillation counting (LSC) is a direct measurement of alpha radionuclides using the pulse shape discrimination technique (PSA) provided by a specific LSA [84,128]. This method involves placing the sample solution in a glass or plastic scintillation bottle, adding a scintillation cocktail to configure it, and placing it in a liquid flash machine for testing [129]. Liquid scintillation counting has been used by some scholars to detect ^210^Po content in samples [103]. Its main advantage is that it eliminates contamination in the detector chamber and also has a high efficiency in detecting alpha particles [28,129]. However, the unoptimized LSC method has poor energy resolution and high detection limits, and the detection limit for the determination of ^210^Po is an order of magnitude higher than that of alpha spectrometry. The presence of other alpha emitters and beta counting is critical to interfering with the measurement. In addition, chemical yield monitoring using ^209^Po or other radioisotopes of polonium does not work in this case [128]. This method is therefore not very useful for measuring the activity of individual alpha-emitting nuclides [5]. Its observed lower detection limit was obtained by scholars who optimized recovery, efficiency, and background counting at ^210^Po [130,131].

The LSC method is also commonly used by researchers for the simultaneous determination of ^210^Po and ^210^Pb in samples [128,129]. In the case of simultaneous detection of ^210^Po and ^210^Pb, determining the appropriate PSA value is important to avoid misclassification of alpha and beta pulses. The study by Ozden et al. concluded that the optimal PSA level for ^210^Po is “10” [129]. In addition, in order to achieve satisfactory alpha/beta screening, the ratio of beta to alpha activity in the sample must not be too high, and for alpha activity measurement, a radioactivity ratio of β to α of less than 100 is generally required [132]. The LSC method avoids the step of separating polonium from the sample and uses alpha spectrometry for detection, saving experimental time and eliminating the cumbersome step of sample preparation.

As one of the LSC, photonic electron rejection alpha liquid scintillation (PERALS^®^) spectroscopy is an emerging method for determining ^210^Po in environmental samples by combining the chemical separation of liquid-liquid extraction with the measurement of α activity in water-immiscible scintillators [28]. Compared with alpha spectrometry, alpha liquid scintillation with β/γ rejection (PERALS^®^) sample preparation is both fast and sensitive and can be used for the detection of native ^210^Po [133]. Case and McDowell applied liquid scintillation to the measurement of polonium-210 in various types of samples, including ore, tailings, and environmental samples [134]. From a 7 M phosphoric acid-0.01 M hydrochloric acid solution, a 0.20 M trioctylphosphine oxide solution (together with scintillators) was extracted in toluene to concentrate polonium-210 and separate it from interfering elements such as iron. Polonium-210 was then measured by counting alpha radiation at 5.3 MeV with a photon/electron suppression alpha liquid scintillation spectrometer [134]. However, the PERALS^®^ system offers poor resolution, and its main interfering nuclides are ^228^Th and ^239^Pu. This means that a separation of ^210^Po from other nuclides is required [28,133]. Previous studies have shown that Polex, TNOA, and TOPO liquid scintillants are also effective in extracting ^210^Po from spiked water samples without the need for ^238^U or ^234^U co-extraction steps [134].

When using the PERALS^®^ method to determine the content of ^210^Po in water, an appropriate amount of phosphoric acid must be added to the pre-enriched water sample, and other acids must be removed by evaporation. After this, one must add tracer, hydrochloric acid, and extraction scintillation solution before shaking well and letting stand while waiting for the extraction to complete. After the organic layer and the water layer are clearly separated, the transparent liquid of the organic phase layer can be reserved, and argon gas can then be sprayed on the counting tube to prevent oxidation from affecting the final sample. The final processed sample can then be placed on a liquid scintillation counter. Although the detection limit for liquid scintillation alpha counting measurements is low, the solution containing ^210^Po still needs to be transparent enough for light collection [84]. In addition, the PERALS^®^ method requires fewer water samples, can allow direct measurement without complex chemical treatment, and is more convenient to operate than other methods. The main disadvantage of PERALS^®^, however, is that commercial-grade phosphoric acid contains a large amount of ^210^Pb and ^210^Po as impurities, so high-purity-grade phosphoric acid [28] is required.

### 5.3. Large-Area Screen Grid Spectrometry

The main step of the large-area screen grid ionization chamber measurement method is to put the pretreated sample into a beaker, add distilled water, and then use an ultrasonic cleaner to crush the sample into about 1 micron particles. Then, the sample is spread in a vacuum drying oven to make a large-area sample source [84,135]. This prepared sample source can then be measured in a large-area α grid ionization chamber. After microwave digestion of 0.5 g of seafood, Li et al. [136] used large-area screen grid spectrometry and found that the detection efficiency of ^210^Po was about 30%. The advantage of this method is that the detection source area is large, and a lower detection limit can thus be obtained [136]. Furthermore, the sample preparation process is simple, the sample is not easy to cross-contaminate, it loses almost no nuclides, and the total α in the sample and the activity of more than 10 α radionuclides, including ^210^Po, can be measured at the same time [127]. However, the requirements for the particles after the sample crushing and grinding treatment are high: the diameter needs to be below 1 micron, and the particle uniformity must be good enough [84]. At present, this method has not been widely used because of the instability of the thickness, uniformity, and firmness of the large-area sources.

## 6. Domestic and International Standards

A series of standard determination methods have been published for ^210^Po in environmental and food samples. The standard methods for ^210^Po in environmental and food samples are listed in Table 2. According to the detection methods used in domestic and international standards, alpha spectroscopy is the most widely used method, which involves the most extensive types of samples tested and has had the most studies carried out on it. Therefore, its determination technology is the most mature.

## 7. Conclusions and Perspectives

This article reviewed the analytical methods for detecting ^210^Po in environmental media and biological samples that have been in use since the 21st century and summarized the advantages and disadvantages of these current sample preparation, chemical separation, sedimentation source preparation, and ^210^Po detection methods. At present, pretreatment methods primarily include evaporation, co-precipitation, chelation, wet digestion, and microwave digestion. The most widely used of these is wet digestion, which efficiently dissolves organic matter in samples at lower temperatures with a fast digestion process and low loss of elements. In addition, many new methods have emerged, such as microwave acid digestion, which also shows good decomposition efficiency. Furthermore, crown ether extraction chromatography is gradually replacing solvent extraction due to its high resolution. Among the various methods for measuring ^210^Po, alpha spectrometry is widely regarded as the most practical analytical technique for quantifying radioactive polonium isotopes. Photonic electron rejection in liquid alpha spectroscopy is also gaining traction due to its simple and rapid sample preparation. Liquid scintillation counting and large-area screen grid spectroscopy, on the other hand, are less commonly used by researchers, mainly because of the high pre-treatment requirements.

Traditional detection methods have complex pretreatment steps, and the whole experiment typically takes a long time, which is not suitable for rapid detection during radiological emergencies. At present, the main bottleneck is how to reach the detection limit of trace ^210^Po in environmental and biological samples through quick and convenient pretreatment. Some literature reports that polonium can be automatically deposited in dilute hydrochloric acid systems containing ascorbic acid, hydroxylamine hydrochloride, etc., without any prior chemical separation, but this is only applicable to individual samples (e.g., biological samples, sediments), and other researchers have raised the difficulties of this approach. Future research will focus on improving the efficiency of the pretreatment, chemical separation, and source preparation stages. In terms of pretreatment, rapid treatment of a large number of insoluble samples needs to be explored. Although the preparation of the photon electron-rejecting liquid alpha spectroscopy is simple and fast, the phosphoric acid used in this method is often contaminated with ^210^Pb and ^210^Po impurities. The main improvement in alpha spectroscopy is to avoid contamination of the detector with tracer without extending the time interval between source preparation and determination. Therefore, urgent research is needed into a fast, efficient, and low-pollution alternative to the traditional method.

## Figures and Tables

**Figure 1 molecules-28-06268-f001:**
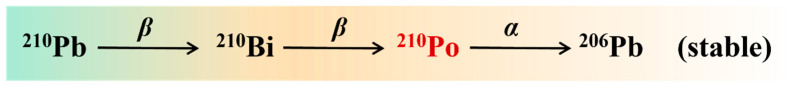
The decay chain of ^210^Po [11].

**Figure 2 molecules-28-06268-f002:**
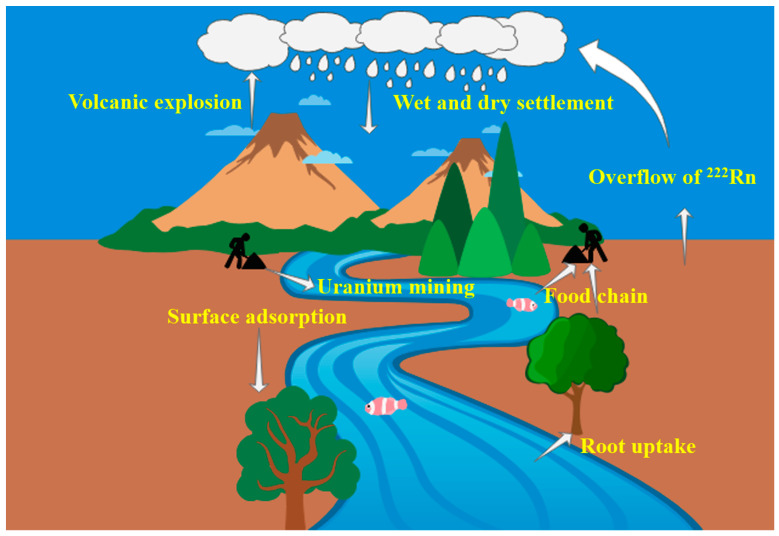
Sources of ^210^Po in the environment and its life cycle in humans and the environment.

**Figure 3 molecules-28-06268-f003:**
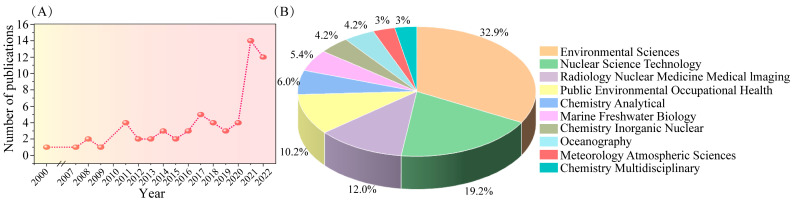
(**A**) Number of publications on ^210^Po from 2000 to 2022. (**B**) Visualization of data for the main application areas of ^210^Po. Data extracted from Web of Science (April 2023).

**Figure 4 molecules-28-06268-f004:**
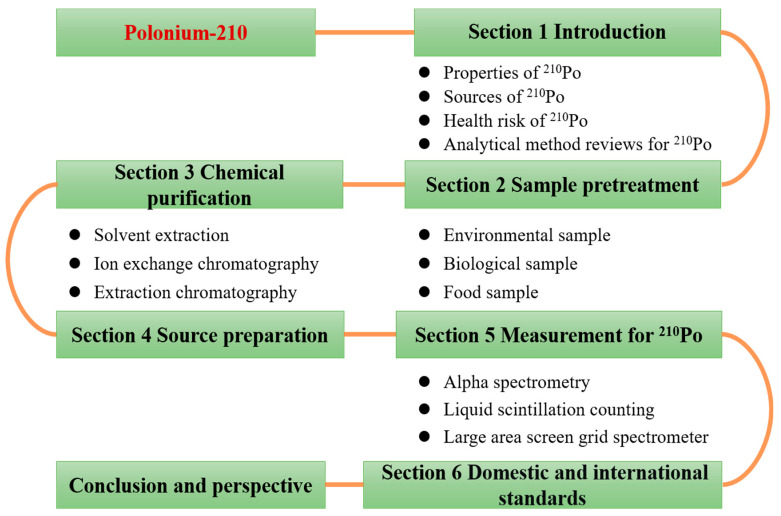
An overview of the structure of the review, including introduction (Section 1), sample preparation (Section 2), chemical purification (Section 3), source preparation (Section 4), measurements (Section 5), domestic and international standards (Section 6), and conclusion and perspective.

**Figure 5 molecules-28-06268-f005:**
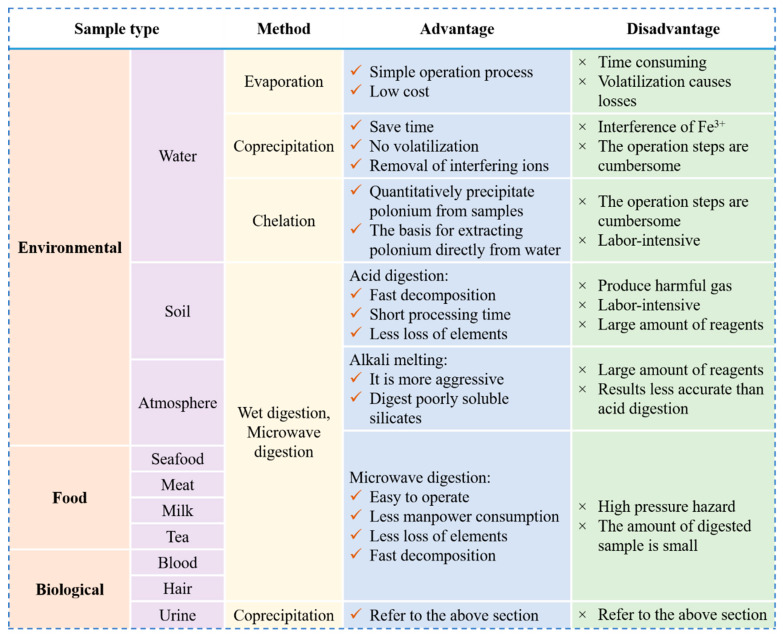
The advantages and disadvantages of the different methods used to pretreat samples.

**Table 1 molecules-28-06268-t001:** Overview of analytical methods developed for determining the quantity of ^210^Po in environmental and biological samples since 2000.

Sample Type	Quantity	Analytical Protocol		Measurement	Detection Limit	Reference
		Pretreatment	Purification			
Fish	20 g (wet weight)	Acid-digestion with HNO_3_ + H_2_O_2_	Sr-resin	α spectrometry	0.1 Bq/kg	[18]
Water	1–2 L	Acid-digestion	Liquid-liquid extraction with Polex™, available from ETRAC™ (Oak Ridge, Tennessee, USA)	liquid scintillation counter	0.4 mBq/L	[20]
Sediment, soil	1–3 g	Acid-digestion	Sr-resin from EIChrom Industries	α spectrometry	/	[29]
Reagent	500 g mineral acid, 50 mL phosphoric acid	Evaporate	Sr.Spec resin from EIChrom Industries	α spectrometry	/	[30]
Urine	100–200 mL	Acid-digestion with conc. HNO_3_ + HCl	MnO_2_ coprecipitation	α spectrometry	1.5 mBq/d	[1]
Seabirds	/	Acid-digestion	Dowex 1-X8 resin	α spectrometry	/	[8]
Fish tissues	0.5–1 g	Acid-digestion	/	α spectrometry	/	[24]
Mussels	0.5–3.5 g	Acid-digestion	/	α spectrometry	/	[19]
Water, urine	10 mL	Water: filtration, urine: Oxidation with KBrO_3_ and Centrifuge	Copper sulfide micro-precipitation	α spectrometry	Water: 0.04 Bq/L Urine: 0.05 Bq/L	[21]
Sludge	0.9 g	Microwave digestion	PS resin from Department of Chemical Engineering and Analytical Chemistry of the University of Barcelona	α spectrometry	0.5 Bq/kg	[31]
Air filters	/	Alkaline fusion	Iron hydroxide co-precipitation and *N,N,N′,N′*-tetra-n-octyldiglycolamide (DGA) resin from Eichrom Technologies, LLC, (Lisle, Illinois, USA) and Triskem International (Bruz, France)	α spectrometry	700 uBq	[32]
Cigarettes	0.2–4.9 g	Acid-digestion	/	α spectrometry	1 mBq/sample	[33]
Medical plants	50–600 g	Acid-digestion	/	α spectrometry	0.1 Bq/kg_dry_	[34]
Matrices	1 g	Alkaline fusion	Iron hydroxide co-precipitation and DGA resin from Eichrom Technologies, LLC, (Lisle, Illinois, USA) and Triskem International (Bruz, France)	α spectrometry	1.4 mBq/g (4 h) 700 uBq/g (12 h)	[35]
Mussels	0.5 g	Microwave digestion	/	α spectrometry	/	[23]
Marine birds	1–5 kg	Acid-digestion	/	α spectrometry	/	[9]
Cigarette and tobacco crops	3–4 g	Acid-digestion	/	α spectrometry	2 Bq/kg	[36]
Seafood, terrestrial, sediment	100 g	Acid-digestion	Filtration	α spectrometry	/	[37]
Water	390 mL	Acid-digestion	Liquid-liquid Extraction	α spectrometry	3.5 mBq/L	[38]
Sediment, pore water	Sediment: /pore water: 94–284 mL	Acid-digestion	Extractions with N_2_	α spectrometry	/	[39]
Urine	/	Evaporate, acid-digestion with conc. HCl + H_2_O_2_	Filtration	α spectrometry	/	[40]
Water	300 mL	Acid-digestion with conc. HCl + H_2_O_2_	TRU resin from Eichrom Technologies (Lisle, USA), Sr resin, Ra resin from BioRad (part #7324661, Mississauga, Canada)	α spectrometry	23 mBq/kg	[41]
Mining residues	0.1 g	Acid-digestion with conc. HCl + HNO_3_	TRU resin, Sr resin from Eichrom Technologies (Lisle, USA)	α spectrometry	/	[42]
Food	5 g	Acid-digestion with HNO_3_ + H_2_O_2_	Filtration	α spectrometry	/	[43]
Poultry	1 g	Acid-digestion	/	α spectrometry	/	[44]
Water	1 L/2 L	Calcium phosphate co-precipitation	Bismuth phosphate micro-precipitation	α spectrometry	200 mL: 6.3 mBq/L 1 L: 0.4 mBq/L	[45]
Water, soil	Water: 4 L, soil: 200 mg	Water: [Fe(OH)_3_] co-precipitation, soil: microwave digestion	/	α spectrometry	Water:0.1 mBq/L Soil:2 Bq/kg	[46]
blood	10 mL	Acid-digestion with conc. HNO_3_	/	α spectrometry	0.1 Bq	[47]

**Table 2 molecules-28-06268-t002:** Standard method for the determination of ^210^Po in environmental and food samples at home and abroad.

Standard Number	Standard Name	Brief Introduction	Issuing Agency, Release Year	Detection Limit	Country	Reference
GB 14883.5-2016	Determination of the radioactive substances ^210^Po in food	Self-deposition on silver or nickel slices—alpha spectrometry	National Health and Family Planning Commission (PRC), 2016	0.74 Bq/g (dry)	China	[109]
HJ 813-2016	Analysis of Polonium-210 in water	Self-deposited on silver plates—alpha spectrometry	Ministry of Ecology and Environment of the People’s Republic of China, 2016	1 mBq/L	China	[110]
ISO/DIS 13161:2020	Water quality—Polonium 210—Test method using alpha spectrometry	Self-deposition—alpha spectrometry	International Organization for Standardization (ISO), 2020	5 mBq/L	Switzerland	[137]
IAEA/AQ/12	A Procedure for the Determination of Po-210 in Water Samples by Alpha Spectrometry	Solvent extraction method, or Sr resin extraction chromatography separation -α spectrometer measurement	International Atomic Energy Agency (IAEA), 2009	2 mBq/L	Austria	[138]
IAEA/AQ/34	A Procedure for the Sequential Determination of Radionuclides in Phosphogypsum	Chromatographic separation of crown ether extraction and determination of polonium-210 by self-deposition method	International Atomic Energy Agency (IAEA), 2014	1.2 Bq/kg	Austria	[139]

## Data Availability

Not applicable.

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
