# Peer review of "Detection of Polonium-210 in Environmental, Biological and Food Samples: A Review"

_molecules, 2023, doi:10.3390/molecules28176268_

Round 1
Reviewer 1 Report
This manuscript present a comprehensive review of the analytical method for the determination of 210Po in different biological and environmental samples, which is useful for the scientists who are new comer to this field, and aims to establish a method for the determination of 210Po. It is therefore worth to be published.
Specific comments:
1. It will be even valuable if the author could compare different methods for sample preparation, separation and measurement of 210Po, instead of just pile up all available methods, which will be benefit to the reader, especially beginner for their choice in the establishment of the method in their lab.
2. The decomposition methods of sample is the most critical step in the determination of 210Po, because of the volatality property of Po, It will be helpful if this issue and behind principle was presented and discussed for a better understanding of the reader for these methods.
3. As an alpha emitting radionuclide, 210Po is often measured by alpha spectrometry after prepared in a thin target. LSC can be also used for its measurement with a high counting efficiency, but the interference of other alpha emitters and beta counting are critical interference. In addition, the chemical yield monitoring using 209Po or other radioisotopes of Po does not works in this case. All these issues are important and need to be well presented.
4. The so-called PERALS measurement technique is one of LSC, and often used for measurement of alpha emitting radionuclides in LSC, i.e. discrimination of alpha and beta emission. While, it was presented as another technique, which might be improved.
Author Response
Response to Reviewer 1:
Recommendation: Minor revisions needed as noted.
Comments: This manuscript present a comprehensive review of the analytical method for the determination of 210Po in different biological and environmental samples, which is useful for the scientists who are new comer to this field, and aims to establish a method for the determination of 210Po. It is therefore worth to be published.
---We thank the reviewer for the comments, and have made the revision as shown below.
- It will be even valuable if the author could compare different methods for sample preparation, separation and measurement of 210Po, instead of just pile up all available methods, which will be benefit to the reader, especially beginner for their choice in the establishment of the method in their lab.
--- We thank the reviewer for this suggestion. In our original manuscript, we have compared the advantages and disadvantages of different sample pretreatment methods.
Line 318, Page 9: “The advantages and disadvantages of all of the above pretreatment methods for environmental, biological, and food samples are detailed in Figure 5.”
Figure 5. The advantages and disadvantages of the different methods used to pretreat samples.
We have added a comparison of different methods for separation, source preparation and measurement of 210Po to the manuscript and summarized the advantages and disadvantages of each method in graphical form, which will be benefit to the reader, especially beginner for their choice in the establishment of the method in their lab. Please check them as below:
Line 494, Page 13: “However, it has been demonstrated that the reduction of Te(IV) to monomeric Te is in-complete in acidic systems in ascorbic acid and not in hydroxylamine hydrochloride. This phenomenon resulted in a very low recovery of polonium, almost close to zero[28].”
Figure S1. The advantages and disadvantages of the different chemical purification for 210Po.
Figure S2. The advantages and disadvantages of the different source preparation for 210Po.
Figure S3. The advantages and disadvantages of the different radioactivity measurements for 210Po.
- The decomposition methods of sample is the most critical step in the determination of 210Po, because of the volatality property of Po, it will be helpful if this issue and behind principle was presented and discussed for a better understanding of the reader for these methods.
--- We thank the reviewer for this suggestion. According to the reported literature (J. Radiat. Res. Appl. Sc., 2015, 8, 590-596; Radiochim. Acta, 2018, 106, 787-792), Since volatile polonium can escape, the radiochemical yield is significantly reduced. The time required for processing can be reduced by different decomposition methods while achieving high yields. Therefore, the decomposition method of sample is the most critical step in the determination of 210Po. The related revision was added in the text. Please check them as below:
Line 112, Page 5: The original sentence “The most crucial aspect during pretreatment is temperature control to prevent polonium loss caused by volatilization at high temperatures” was revised to “The most crucial aspect during pretreatment is temperature control, which prevents polonium from volatilizing at high temperatures and resulting in significantly reduced radiochemical yield[2, 48].”
Line 119, Page 6: The original sentence “…temperature control is essential during pretreatment of both environmental and biological samples and for brevity is not specifically noted below” was revised to “…temperature control is essential during pretreatment of both environmental and biological samples. In addition, the time required for processing can be reduced by different pretreatment methods while achieving high yields.”
- As an alpha emitting radionuclide, 210Po is often measured by alpha spectrometry after prepared in a thin target. LSC can be also used for its measurement with a high counting efficiency, but the interference of other alpha emitters and beta counting are critical interference. In addition, the chemical yield monitoring using 209Po or other radioisotopes of Po does not work in this case. All these issues are important and need to be well presented.
--- We thank the reviewer for this comment, and we have already mentioned the high counting efficiency of LSC measurement in detecting α particles and the shortcomings of this method in our manuscript.
Line 550, Page 15: The original sentence “…has a high efficiency in detecting alpha particles[26, 126]. However, the unoptimized LSC method has poor energy resolution and high detection limits, and the detection limit for the determination of 210Po is an order of magnitude higher than that of alpha spectrometry.”
Based on the suggestions of the reviewers, we have added some related contents in our paper. Please check them as below:
Line 552, Page 15: “…higher than that of alpha spectrometry. The presence of other alpha emitters and beta counting is critical to interfering with the measurement. In addition, the chemical yield monitoring using 209Po or other radioisotopes of polonium does not work in this case[128].”
[128] Kong X, Yin L, Ji Y. Simultaneous determination of 210Pb and 210Po in seafood samples using liquid scintillation counting. Journal of Environmental Radioactivity. 2021, 231: 106553.
Figure 4. An overview of the structure of the review, including introduction (section 1), sample preparation (section 2), chemical purification (section 3), source preparation (section 4), measurements (section 5), domestic and for international standards (section 6) and conclusion and perspective.
- The so-called PERALS measurement technique is one of LSC, and often used for measurement of alpha emitting radionuclides in LSC, i.e. discrimination of alpha and beta emission. While, it was presented as another technique, which might be improved.
--- We thank the reviewer for this suggestion. The PERALS measurement technique is one of LSC and often used for measurement of alpha emitting radionuclides in LSC, i.e. discrimination of alpha and beta emission. We have integrated the relevant content in sections 5.2 and 5.3 in newly submitted manuscript. The related words were revised and added into the text. Please check them as below:
Line 510, Page 14: The original sentence “There are four main methods for the detection of 210Po in environmental and bio-logical media: alpha spectrometry, liquid scintillation counting, photon electron re-jecting alpha liquid scintillating spectroscopy and large area screen grid spectrometry. We summarize the principles, advantages, and disadvantages of each method in detail in the following subsections” were revised to “There are three main methods for the detection of 210Po in environmental and biological media: alpha spectrometry, liquid scintillation counting and large area screen grid spectrometry. The principles, advantages, and disadvantages of each method are summarized in detail in the following subsections and Figure S3”.
Line 569, Page 15, “As one of LSC, photonic electron rejection alpha liquid scintillation…”

Reviewer 2 Report
The topic is very interesting and the text is very well written and organized. It discusses a comprehensive review about 210Po extraction and analysis in Environmental, Biological and Food Samples. Many illustrating figures are well represented in addition to summarizing tables. However, minor revision is required.
1. In the title “ write full name for polonium-210 “
2. The following articles should be cited in the introduction
Polonium-210 poisoning: a first-hand account, https://www.thelancet.com/journals/lancet/article/PIIS0140-6736(16)00144-6/fulltext
Determination of polonium - 210 (210Po) in food and water: a review (2014-2019) https://inis.iaea.org/search/search.aspx?orig_q=RN:53002370
3. Figure (3 A) resolution should be improved, I suggest to be enlarged to fit the screen width to be easily read.
4. Fig 4 could be used as a graphical abstract.
5. In line 189, DGA full name should be mentioned in the first mentioning.
6. Line 325, correct typo error. [interfered with by other ions,]
7. A figure for item 5. For the four main types of Radioactivity measurements will be very useful as a summary. Additionally, diagram for each item will be helpful guide, they can be added for the supplementary file
8. Reference 134 should be written with full details.
9. A new column should be added for table 2 about origin country for the code.
10. Name of the country should be added for the acknowledgment
Best wishes
Author Response
Response to Reviewer 2:
Recommendation: Minor revisions needed as noted.
Comments: The topic is very interesting and the text is very well written and organized. It discusses a comprehensive review about 210Po extraction and analysis in Environmental, Biological and Food Samples. However, minor revision is required.
---We thank the reviewer for the comments, and have made the revision as shown below.
- In the title “write full name for polonium-210 ”.
--- We thank the reviewer for this suggestion. Line 2 " 210Po " in the tittle has been revised to be " polonium-210 ".
- The following articles should be cited in the introduction Polonium-210 poisoning: a first-hand account, https://www.thelancet.com/journals/lancet/article/PIIS0140-6736(16)00144-6/fulltext
Determination of polonium - 210 (210Po) in food and water: a review (2014-2019) https://inis.iaea.org/search/search.aspx?orig_q=RN:53002370
--- We thank the reviewer for the suggestion to improve our paper. These references have been added into our manuscript, please check them as follows:
[5] Nathwani, A.C.; Down, J.F.; Goldstone, J.; Yassin, J.; Dargan, P.I.; Virchis, A.; Gent, N.; Lloyd, D.; Harrison, J.D. Polonium-210 poisoning: a first-hand account. Lancet. 2016, 388, 1075-1080.
[7] Barbosa Gonzalez, N.R.; Ramos Rincon, J.M. Determination of polonium - 210 (210Po) in food and water: a review (2014-2019). Revista Investigaciones Y Aplicaciones Nucleares (Online). 2021, 5, 26-43.
- Figure 3A resolution should be improved, I suggest to be enlarged to fit the screen width to be easily read.
--- We thank the reviewer for the suggestion. We have revised the figures accordingly. Please check it as below:
Figure 3. (A) Number of publications on 210Po from 2000 to 2022. (B) Visualization of data for the main application areas of 210Po. Data extracted from Web of Science (April 2023).
- Fig 4 could be used as a graphical abstract.
--- We thank the reviewer for this suggestion. Fig.4 has been used as a graphical abstract for this paper.
- In line 189, DGA full name should be mentioned in the first mentioning.
--- We thank the reviewer for this suggestion. The full name of N, N, N’, N’-tetra-n-octyldiglycolamide (DGA) has been used since it was first mentioned in Table 1 of this paper. In line 164, the full name of the DGA is also mentioned.
Line 164, Page 6: “…co-precipitation for sample pretreatment, separation and purification using N,N,N’,N’-tetra-n-octyldiglycolamide (DGA) resin…”
- Line 325, correct typo error. [interfered with by other ions,]
--- We thank the reviewer for this comment, and we apologize for the mistake. We have revised it, please check it as below:
Line 327, “…interfered with by other ions” was revised to “…affected by other ions.”
- A figure for item 5. For the four main types of Radioactivity measurements will be very useful as a summary. Additionally, diagram for each item will be helpful guide, they can be added for the supplementary file.
--- We thank the reviewer for this suggestion. We have summarized the radioactivity measurements described in our manuscript in graphical form and added them to the supplementary file. The related words were revised and added into the text. Please check them as below:
Line 510, Page 14: The original sentence “There are four main methods for the detection of 210Po in environmental and bio-logical media: alpha spectrometry, liquid scintillation counting, photon electron rejecting alpha liquid scintillating spectroscopy and large area screen grid spectrometry. We summarize the principles, advantages, and disadvantages of each method in detail in the following subsections” were revised to “There are three main methods for the detection of 210Po in environmental and biological media: alpha spectrometry, liquid scintillation counting and large area screen grid spectrometry. The principles, advantages, and disadvantages of each method are summarized in detail in the following subsections and Figure S3”.
Line 660, Page 17: “Supplementary Materials: The supporting information including the advantages and disadvantages of the different sample preparation, separation and measurement, source preparation and radioactivity measurements for 210Po.”
Figure S3. The advantages and disadvantages of the different radioactivity measurements for 210Po.
- Reference 134 should be written with full details.
--- We thank the reviewer for the suggestion to improve our paper. Reference 134 in our original manuscript has been written with full details in our paper. Please check it as follows:
[137] ISO/DIS 13161: 2020, Water quality-Polonium 210-Test method using alpha spectrometry. International Standard Organization: Geneva, Switzerland, 2020.
- A new column should be added for table 2 about origin country for the code.
--- We thank the reviewer for this suggestion. We have added the original country for the code in table 2. Please check them in newly submitted manuscript.
- Name of the country should be added for the acknowledgment.
--- We thank the reviewer for this suggestion. We have added the name of the country to the acknowledgment. Please check them as below:
Line 657, Page 17: “The authors gratefully acknowledge financial support from the Zhejiang Health Science and Technology Plan (China, No. 2021KY613, 2022RC120, 2022KY130, 2022KY132, 2023KY643), Project of South Zhejiang Institute of Radiation Medicine and Nuclear Technology (China, No. ZFY-2021-K-003, ZFY-2022-K-001, ZFY-2022-K-006).”
